# Frequent gamblers' perceptions of the role of gambling marketing in their behaviour: An interpretative phenomenological analysis

**Scott Houghton** [1]*, **Georgia Punton**[1], **Emma Casey**[2], **Andrew McNeill**[1,3], **Mark Moss**[1]

**1** Department of Psychology, Northumbria University at Newcastle, Newcastle-upon-Tyne, United Kingdom, **2** Department of Sociology, University of York, York, United Kingdom, **3** Psychology and Communication Technology (PaCT) Lab, Northumbria University, Newcastle-upon-Tyne, United Kingdom

* scott2.houghton@northumbria.ac.uk

## Abstract

This study explored how frequent gamblers perceive gambling marketing and the role they feel it has in their gambling behaviour. Ten frequent gamblers participated in semi-structured interviews oriented around their experiences of gambling marketing. An interpretative phenomenological analysis of the data led to three overarching themes: exploiting gambling marketing for personal gain; gambling marketing as a test of self-control; and safer gambling messages marketing perceived as ineffective. These themes encapsulated participants' views of gambling marketing as something they could take advantage of to increase their own gambling success. Marketing was also perceived as a test of self-control among self-identified experienced gamblers, although identified as a risk to those who are considered more vulnerable. Finally, safer gambling messages included within marketing was considered ineffective due to perceived insincerity and being seen as an 'afterthought' by marketers. In support of previous research, the current investigation highlights concerning narratives around self-control and perceived risk, as encapsulated within gambling marketing, and these are evident in the perceptions of frequent gamblers. Given gamblers' perceived lack of effectiveness of current safer gambling messages within marketing, future research should explore new avenues for safer gambling promotion.

## Introduction

A recent review examining the social and economic cost of gambling-related harms in Great Britain estimated the annual cost to be £1.27 billion, although this is likely underestimated due to a lack of high-quality research quantifying such harms [1]. Gambling-related harms span across numerous domains, from financial impacts and emotional distress to lesser considered factors such as reduced work performance and cultural harms [2]. Harms do not only impact the individual gambler, with a recent literature review highlighting the significant negative effects gambling can have on concerned significant others [3]. Given the wide-ranging impact of gambling-related harm at both an individual and population level, much focus within the literature has been dedicated to assessing factors that contribute towards such harm.

**Data Availability Statement:** Data cannot be shared publicly as it is qualitative data on a sensitive topic area that could be personally identifiable based on the depth of discussion

present within the interviews. Additionally, when ethical approval was sought for the current research, participants were not told that their transcripts would be made available online. However, a full analytical trail is available at https://osf.io/utzds/, which documents the analysis process from initial coding to final themes.

**Funding:** The study was carried out as a part of the lead authors' (SH) PhD studies. The PhD was funded by GambleAware (https://www.begambleaware.org/). The funder played no role in the design, data collection, data analysis, decision to publish or the preparation of the manuscript.

**Competing interests:** The study was carried out as a part of the lead authors' (SH) PhD studies. The PhD was funded by GambleAware. No other author has competing interests to declare.

One such proposed factor is gambling marketing which is shown to normalise gambling within sporting environments [4, 5], influence children to gamble [6], encourage individuals to gamble when they are aiming to quit [7] and encourage individuals to place more risky bets [8]. The gambling industry also invests a lot of money into marketing, with £8.3 million a week spent on paid for television advertising [9]. As such, it is important that research explores the role marketing has upon individuals' behaviour and the potential to cause harm. However, assessing the impact of marketing upon gambling behaviour proves to be methodologically challenging due to the wide-ranging marketing strategies employed [6] and the fact that marketing is just one of many factors that could influence gambling behaviour. Whilst carefully designed experimental studies can demonstrate how specific aspects of marketing may be harmful to gamblers within a simulated gambling environment, there are questions over whether such findings would replicate in a real-world setting. There has some encouraging success with longitudinal research highlighting increased spending related to increased exposure to gambling marketing, as well as the most influential types of marketing inducements [10]. However, to develop a more holistic understanding of the role marketing has within an individual's gambling behaviour, it is important to understand how marketing is perceived by bettors.

Despite this, there are only a limited number of studies which have explored the perceived impact of gambling marketing qualitatively. For example, semi-structured interviews with 25 Swedish disordered gamblers found advertising was perceived to increase their gambling problems, due to triggering impulses to gamble and creating difficulties in following through on a decision to cease gambling [7]. More recently, focus groups with 43 treatment-seeking disordered gamblers in Spain explored themes around the perceived impact of marketing on their gambling behaviour [11]; participants found price-related gambling promotions to be particularly persuasive by prompting them to calculate the potential advantage to be gained in accepting such offers. Additionally, advertising was stated to be effective in instigating the uptake of a new gambling product and also caused anticipatory anxiety in situations where gamblers expected to encounter gambling advertising.

However, these studies only focus on how marketing influences those who have been diagnosed with gambling disorder. Considering gambling-related harms occur across the spectrum of problematic gambling behaviour [12] and research demonstrating that marketing can prompt riskier gambling behaviour regardless of problem gambling levels [13], more recent research has explored attitudes and opinions towards marketing in those who gamble frequently but do not have a diagnosis of gambling disorder. Recent qualitative studies in the UK have highlighted a range of key findings on perceptions of gambling marketing from such populations: including the frequency at which individuals report seeing gambling advertising, its normalising effect within society, the attractiveness of inducements within gambling and the lack of effectiveness of safer gambling messaging [14, 15]. A grounded theory approach was also taken in an Australian study and highlighted the key component of betting-related responses to wagering inducements was through using them to minimize their losses [16]. Together, these studies demonstrate that taking a qualitative approach allows gamblers to expand on how they view gambling marketing and how it might affect their gambling behaviour. However, the experience of gambling and the general relationship between gamblers and gambling marketing strategies is unique to each individual, and highly context specific [17], and so, it is important to encapsulate these individual nuances through idiographic, qualitative means [18].

One qualitative approach that is useful in understanding how individuals interpret experiences within their lives is Interpretative Phenomenological Analysis (IPA). The phenomenological nature of IPA ensures that such research revolves around developing an understanding

of how people individually make sense of their social world and its experiences; [19]. As such, IPA is a commonly employed method within gambling research [20–23], due to its focus on the individual as well as the wider homogenous population experiencing certain phenomena [19]. Within the current study, the use of IPA will not objectively measure the impact of gambling marketing upon behaviour, but instead explore how frequent gamblers make sense of the role marketing plays within their gambling lives. This is important to consider because the impact of more subtle marketing strategies [24] is not objectively measurable at the individual level and therefore encouraging reflection upon such strategies is key to understanding how gamblers perceive and engage with them. For example, exploiting probability biases in gamblers' behaviour may increase the amount of money spent by gamblers, but the gambler still engages in a conscious decision (accessible to phenomenological reflection) that the advertised odds look better than an alternative. It is also difficult to untangle the impact of other marketing strategies due to their highly integrative nature. Whilst such methods may be criticised from a positivist viewpoint as being too subjective, interpretation is actively encouraged within IPA as a means of exploring feelings, emotions and meanings [25]. Furthermore, the experience of marketing on gamblers *is* inherently subjective, meaning that the only way to adequately account for it is to explore gamblers' personal accounts of such phenomena.

Therefore, the aim of the current study is to use IPA to answer the research question '*how do frequent gamblers perceive gambling marketing and make sense of its role in their behaviour*?'.

## Method

### Approach

The current study will utilise a qualitative approach which will flexibly follow the IPA methodology. IPA in the current context aligns with a critical realist approach [26]; realist insofar as we hold that participants experience real effects of marketing on their thinking and behaviour which can be recounted; critical insofar as the explanations of such experiences are likely affected by self-presentation concerns. The phenomenological aspect of IPA is also valuable in gambling research [27], as it is important to explore and understand how certain phenomena are personally experienced and understood. IPA is idiographic in nature and is therefore useful when exploring complex social phenomena by identifying individual idiosyncrasies in behaviour and experience [28]. IPA is therefore considered more appropriate than other qualitative approaches, mainly due to its dual-analytic approach, which focuses on idiographic accounts, as well as an overall patterning of meaning across participant accounts [19, 29].

### Participants

Ten participants were recruited to take part in the study via purposeful sampling, as is typical in IPA [19]. Emphasis was placed on the importance of recruiting participants that offer a detailed insight into a particular experience, and therefore a homogenous sample was sought [19]. The size of the sample is deemed less important in IPA, with the concept of data saturation argued to be problematic within such an idiographic analysis [30]. Sample size was instead guided by recommendations for doctoral IPA research from leading researchers in the area [19]. The inclusion criteria for taking part in the study was that participants had to be frequent gamblers between the age of 18 and 34, as individuals from this group report gambling mostly in response to marketing [31].

In the absence of a consistent definition within the literature [32–34], frequent gambling was classified as gambling on three days a week or more. Several recruitment strategies were employed for this investigation, including advertisements on a university campus, on social

**Table 1. Participant demographic information.**

| Participant Pseudonym | Age | Sex | Employment Status | Ethnicity | Relationship Status | PGSI Score (Risk) |
|---|---|---|---|---|---|---|
| Tom | 22 | Male | FT student | White-British | Single | 4 (Moderate-risk) |
| Mark | 22 | Male | FT student | White-British | Relationship | 10 (Problem) |
| Harry | 21 | Male | FT student | White-British | Single | 6 (Moderate-risk) |
| Daniel | 22 | Male | FT student | White-British | Single | 3 (Moderate-risk) |
| Connor | 21 | Male | FT student/PT employed | White-British | Relationship | 6 (Moderate-risk) |
| David | 21 | Male | FT student | White-British | Single | 9 (Problem) |
| Chris | 23 | Male | FT student | White-British | Single | 8 (Problem) |
| Paul | 28 | Male | FT student/PT employed | White-British/Mixed | Relationship | 6 (Moderate-risk) |
| Charlotte | 32 | Female | FT employed | White-British | Single | 2 (Low-risk) |
| James | 20 | Male | FT student | White-British | Single | 10 (Problem) |

media, and among local bookmakers. Participants were compensated with a £15 Amazon voucher for taking part in the study.

Table 1 outlines demographic information for all recruited participants, including age, sex, employment status, ethnicity, relationship status and participant scores on the Problem Gambling Severity Index [PGSI] [35].

## Data collection

The study received full ethical approval from the Northumbria University postgraduate research ethics committee (REF—12156). Upon obtaining written informed consent from each participant, semi-structured interviews were used to collect data. Participants were informed they were free to leave the interview at any point, however nobody chose to do so. Interviews were conducted on a university campus in 2019 and 2020 by researcher SH and lasted between 40 and 70 minutes. The main aim of interviews within IPA research is to allow the participant a platform to provide rich and detailed first-person accounts of a phenomena [36]. Such methods can often uncover elements of reality that we are not able to empirically capture, therefore providing ontological depth [37]. This is particularly relevant to the current study given the need to explore gamblers' personal accounts and experience of marketing strategies, especially considering the need to investigate the potential cause-and-effect relationship between marketing and disordered gambling that may exists at this personal level.

The interview schedule (osf.io/8c9jg) consisted of open-ended questions, alongside relevant prompts and examples of gambling marketing that were shown to participants during the interview. The final schedule consisted of 12 questions, and represented four main categories: personal gambling history, television advertisements, social media marketing and gambling within sport. Such categories allowed the researcher to steer the participants towards discussing experiences of certain types of marketing, whilst retaining the flexibility needed to pursue any interesting topics of discussion that arose through the course of the interview.

## Data analysis

The interviews were transcribed verbatim by the lead researcher (SH) and the data was analysed using IPA, following guidelines of lead researchers within the area [19]. The first stages within the analysis process were to actively listen to each interview read the transcript numerous times, and make exploratory notes on any areas of interest within the transcript. Notes made at this stage were a mixture of descriptive comments, which gave accounts of how participants discussed gambling marketing, and more conceptual comments, which explored the

context in which these discussions were embedded. It is argued that this allows for identification of more abstract concepts that assist the researcher in making sense of the participant's lived experience through 'emergent themes', hence representing the 'double hermeneutic' approach involved in IPA [19]. Emergent themes were thus generated on a case-by-case basis from these exploratory notes and were clustered to create a thematic structure for each participant, capturing the different ways individual participants thought about gambling marketing. A short summary of the findings was then written up to revisit at a later stage of the analysis.

This process was repeated for each of the ten interviews carried out. The researcher used the individual summary of findings to uncover commonalities between emergent themes. Within this process, some themes were clustered together to create an overarching theme. Additionally, some themes were collapsed into a pre-existing theme which subsequently became an overarching theme. Within this theme development, it was important to ensure that the themes generated were theoretically distinct from one another. This allowed for a final thematic structure to be produced, which accounted for shared experience whilst also maintaining the idiographic nature of the analysis. A full analytic trail is accessible through the Open Science Framework (osf.io/utzds/).

### Role of the researcher

As previously noted, a key aspect of IPA interviewing is that participant is provided a platform to openly discuss their experiences of a particular phenomenon [19]. Most of the recruited participants were the same sex as the interviewer (SH) and were around a similar age and such methods of peer-interviewing allowed for a rapport to be developed that encouraged detailed experiential responses from participants [38]. The different stages of data analysis, as described previously, were all documented and are available to view online (osf.io/utzds/), allowing for analytical transparency to be established [39]. Finally, a second researcher (GP) reviewed the analysis to ensure findings were grounded in the experiences of participants.

## Results

Three final themes were developed through the analysis that encapsulate how regular gamblers understand the role gambling marketing plays within their behaviour. The first theme 'taking advantage of gambling marketing for personal gain' explains how participants view marketing as something to be exploited for their own personal gain to increase their own chances of winning. The second theme 'gambling marketing as a test of a gambler's self-control' covers how participants feel tempted to gamble by certain types of gambling marketing and how this is particularly risky for those who are vulnerable. The final theme 'safer gambling messages perceived as ineffective' describes the lack of trust and confidence participants feel towards safer gambling messages.

### Theme 1: Exploiting gambling marketing for personal gain

Throughout the interviews, all participants discussed ways that they can take advantage of gambling marketing to enhance their gambling experience or increase their chances of gambling success. As such, marketing was presented as something which, under a skilled and considered approach, could be beneficial to individual gamblers. For example, numerous participants demonstrated how they were able to use marketing offers to reduce the risk associated with their gambling behaviour.

> 'It's about getting as much as you can from as little as possible, erm, so I want to say Betway are offering if you have a £10 bet ante-post–so before the market before Tuesday you'll get a

*£10 free bet for each day of the festival, so something like that. I'll have a tenner on something beforehand that I think is a banker and then I'll have a free £10 bet Tuesday, Wednesday, Thursday, Friday, so it's a race covered everyday cause it's a free bet. Normally I wouldn't probably bet £10 on that race but cause its free it's easier to, it's easier to justify, I don't have to justify it cause it's their money.'*–Harry (male, PGSI score = 6)

In placing an initial qualifying bet, Harry is able to 'earn' a series of free bets to use on the racing festival. Through stating that these free bets will cover a race every day, it is implied that these free bets afford him the opportunity to maintain a certain frequency to his betting over the course of the festival whilst minimising his risk of losing large sums of money. Interestingly, he then states that he does not need to justify his betting behaviour when betting with free bets, suggesting an internal conflict around levels of spending which usually exists when spending actual money on gambling. So, whilst the initial suggestion from Harry is that the uptake of such offers is to cleverly reduce the financial risk associated with his gambling, a more detailed interpretation highlights how free bets alleviate his internal conflicts around levels of spending on gambling whilst continuing to actively gamble. James similarly puts forward the value in taking advantage of free bets offers to reduce financial risk, commenting upon 'the lack of pressure' they feel when betting with free bets. However, when questioned whether their strategy differed when betting with free bets, he explained how free bets led to making riskier gambling choices.

*' I went to Aspers the casino in [RETRACTED] and they give me like a £5 free bet and, er, I remember the dealer had like a, I had like a four and I had 15 off my two cards and I'd usually stay, I wouldn't, I'd just leave it but I just thought it's £5, hit me again like that sort of thing so it changed my, my strategy, I lost my strategy because it was free, like cause I thought I'm not gonna lose like I'm never gonna lose cause like, and then you do lose cause you put another bet on and do the same.'*—James (male, PGSI score = 10)

The increased freedom which accompanies the lack of financial risk when using free bets allows James to take a riskier approach within his game of blackjack. However, changes to his betting strategy continue when he starts to gamble with his own money, something which he attributes to the feeling that he cannot lose whilst betting with the free bet. This indicates that emotional states that are present when betting with free bets, such as a lack of fear of losing money, can extend beyond the period in which they are using the free bet. As such, whilst the initial purpose of engaging with the marketing offer may be to reduce risk, it may result in riskier behaviour in the long term.

Most participants also discussed ways that they took advantage of gambling marketing to increase their chances of making profit when gambling, such as signing up for multiple bookmakers to take advantage of the sign-up offers, exemplified here by Daniel.

*'I do have accounts, but I don't really use them. I use SkyBet as my main one, my go to and then probably PaddyPower's like sort of like another one that I use but apart from them two I don't actively bet with anybody else. I just used to do the join offers and see if I can like rinse, basically rinse them for some money and then just take off, off into the sunset.'*—Daniel (PGSI score = 3)

Despite only betting regularly with two bookmakers, Daniel has signed up to others to make profit from their sign-up offers. Such offers are thus presented as a one-time opportunity for him to exploit marketing to secure profits from bookmakers he wouldn't normally use.

This conveys the idea that sign-up marketing offers can be exploited by savvy gamblers as part of a wider gambling strategy to increase the chances of making money through gambling. David highlights a further example of this strategy, explaining how price boosted bets can increase the value of certain bets to the point where the implied probability of the odds is lower than their perceived likelihood of the event occurring.

> *'Before I even see the odds you know like, I've got like an opinion or whatever it is but some- times you see the odds and you think the odds are really good and then you know when you see a boost you might think, even if I'm not, even if I don't think that's going to happen the odds are you know, they sort of outweigh the sort of chance of it not happening so, it may make me place a bet, especially if I'm sort of on the, you know, on the fence on it I see better odds and I think, oh might as well'*—David (PGSI score = 9)

Through comparing his pre-conceived judgements of how likely a bet is to win against how likely the advertised odds suggest it is to win, David reaches a judgement on whether the bet represents good value. Boosted odds make bets more appealing and push him towards betting on them even if he does not think the bet is likely to win. This therefore depicts the idea that knowledgeable gamblers can search for bets where the odds are in their favour over the book- maker to secure profit over an extended period of time.

### Theme 2: Gambling marketing as a test of self-control

Despite the consensus amongst participants that marketing could be exploited for personal gain, there was also an agreement that marketing acted as a test of their control of their gam- bling behaviour. Most participants interviewed described how marketing acted as temptation or as a reminder to the gambler, with Mark stating that marketing offers draw him back into gambling after deciding to stop due to financial concerns:

> *'I'll get like obviously marketing, like I get texts off like Ladbrokes and Coral and that, like giv- ing me offers and that and, erm, when I stop I sort of find it like I don't ever get like an urge in my head to go and splash a load of money but I sort of think I, I'll do that it's just a bit of fun init and then sort of progressively gets a bit more and more like progressive'*–Mark, (PGSI score = 10)

It is evident that there is a disconnect between Mark's reasons for gambling and his actual behaviour. Whilst acknowledging that his gambling behaviour escalates to levels that he feels uncomfortable with, he struggles to maintain his attempts to stop gambling due to the enjoy- ment he associates with the activity. Marketing is identified as something which leads him to start gambling again after choosing to stop, implying that marketing offers act as temptation by reminding him of the enjoyment that he gets from gambling. Marketing therefore plays a key role in establishing a cycle of behaviour whereby attempts to stop gambling are prevented from being successful by acting as a reminder of the perceived positive aspects of gambling. Numerous participants also discussed how the high frequency of gambling marketing within their everyday lives made it hard for them to switch off from gambling.

> *'I think I touched on it before where they've got like live odds on things so if you're busy watch- ing a game or whatever things like that I quite often, erm, make us actually go put a bet on or at least have a look so I might not put that exact bet on but I might go have a look and see what other bets are on of like a similar ilk.'*—Connor (PGSI score = 6)

Connor explains how seeing marketing during live sport encourages him to bet, even if he is not interested in the specific bet that is being advertised. This indicates an aversion of focus when watching sport from watching for enjoyment to thinking about what bets could be placed on the game by initiating an evaluation of the advertised bet. Marketing therefore acts as a reminder to the participant of the possibility of gambling on the event that they are watching. Additionally, James discussed how marketing schemes used within casinos glamourise gambling and encourage increased spending.

*'Say I was on four points for the month and there was only a few days left I'd probably just want to get up to that tier because I've seen people who are obviously at these tiers and they make them feel like celebrities, like even their drinks come in a fancier glass, erm, they, they get a valet who'll come and take their coat and they come over every five minutes is everything ok and you, your card even looks different just little things, like mine has just got white card and they'll have like a matt black card and it just looks, it's just the whole, it's like fashion isn't it you try and look better and then you feel more important and you spend more money.'—* James (PGSI score = 10)

James explains tiered marketing scheme whereby gamblers receive differing levels of perks based upon their levels of spending. Whilst acknowledging that such marketing schemes aim to keep the gambler spending, he states that he would be tempted to spend more money to reach the higher tiers if he could afford it due to the celebrity-like treatment that the higher tiers receive. This suggests that the temptation provided by such schemes act by playing upon perceptions of self-worth, making higher spending gamblers feel more important and elevating their social status within the gambling environment.

Despite the acknowledgment that marketing acts as temptation to gamble, most participants stated that advertising did not have any serious impacts upon their gambling behaviour. Instead, most of their concerns around marketing were related to those who they saw as having problematic personalities or who were problem gamblers. For example, in response to a question on what makes a gambler vulnerable, something he previously established as a risk factor for marketing, Tom responds that vulnerability refers to a lack of control over spending, particularly for individuals with major financial responsibilities.

*'Maybe someone that's got less control over how much money they're wanting to be putting into their gambling accounts, certainly I know cause obviously still living at home being a student I don't necessarily have a lot of outgoings of my money so, erm, so maybe when I've got more responsibilities like a house and things like that, er, and people that are also in that situation when they maybe don't have as much disposable income to be gambling with I think that's when it can become a little bit more irresponsible they're kind of targeting their adverts at people who need their money for other things.'—* Tom (PGSI score = 4)

This indicates that gambling advertising triggers those who cannot control their gambling behaviour into spending money that they cannot afford to spend. In distancing himself from such a lack of control or financial responsibilities, Tom suggests that any dangers of gambling marketing do not apply to himself. So, whilst marketing serves as temptation to gamble, the severity of the negative impacts that it can have upon behaviour depends upon individual factors to each gambler. This view is supported by Connor, who argues advertising targets individuals with addictive personalities.

*'I do know plenty people that have quite addictive personalities so the more that they are targeted by gambling companies the more potential there is for them to, er, to kind of succumb to that kind of demand I guess, erm, but like I say not really for me personally, but I definitely know people that would be sucked in by those kind of adverts.'*–Connor (PGSI score = 6)

In responding to an advert shown by the interviewer, Connor argues that the advert would not make him gamble but that he knows other people who would be drawn into betting when viewing it. The use of the metaphor 'sucked in by those kind of adverts' presents marketing as a trap designed to lure a specific sub-group of gamblers into gambling. Whilst this acknowledges the dangers of gambling marketing, it places the responsibility of avoiding such negative consequences on each individual gambler. Through describing the limited impact of advertising upon his own behaviour, Connor implies a superior level of self-control over his gambling behaviour compared to other people that he knows. As such, this suggests that the risks of gambling marketing only exist for those who are not in control of their behaviour and cannot resist the temptation to gamble evoked by such marketing.

## Theme 3: Safer gambling messages perceived as ineffective

The final theme developed from the interviews covered the perception amongst most participants that safer gambling messages lack effectiveness. Concerns were expressed over the content of safer gambling messages, both in terms of the lack of useful information and the uneven balance between prompting people to gamble and promoting safer gambling. For example, Harry discussed how current safer gambling messages within marketing campaigns fail to provide gamblers with the relevant information to reduce the risk associated with their gambling behaviour.

*'There isn't many promoted ways of safer gambling, it's just like when you stop having fun stop betting but there isn't like a, this is a way of trying to reduce your risk and like if you've got an addiction how to help yourself without having to go through all that, there isn't really much guidance on starting safe and not waiting until you're five-thousand pound in debt to try and become safe.'*—Harry (PGSI score = 6)

Harry discusses how the 'when the fun stops stop' safer gambling slogan lacks effectiveness due to an absence of practical application. In highlighting examples where more focused and informative safer gambling advice would be useful for gamblers, this emphasises that the complexity of promoting safer gambling cannot be covered by one uniform slogan. This is because the advice needed for someone beginning to gamble is completely different from the help needed for someone who is experiencing harm from gambling. Building upon this, Chris highlighted the lack of focus on safer gambling with gambling marketing.

*'I feel like the whole safe gambling's just, erm, a bit of like a bit of a blanket over it all sort of thing, like you can easily just rub it off straight away and you wouldn't notice that it wasn't there like, I mean the gamble responsibly bit on the end of the advert I, I think you've already targeted someone by giving the odds or the boost or something in the advert I think with that at the end of it you've already hooked them*—Chris (PGSI score = 8)

Chris describes how he sees safer gambling messages within marketing as ineffective as they are usually incorporated at the end of an advert, after the company has advertised an appealing bet or offer. Safer gambling messaging within marketing are therefore seen as an afterthought and, as such, he pays little attention to it. Taken together, both previous extracts highlight that

gamblers struggle to connect with safer gambling messages due to the way safer gambling content is included within marketing campaigns. Additionally, there were also concerns as to the sincerity of safer gambling messages within marketing from gambling operators.

*'I'm not necessarily sure it's the best message coming from the people that are producing it themselves. I think it should be more like a government thing if, or actually made into a law or something like that, erm, because it's so accessible and it just seems mad if the people that are running are it are necessarily the ones responsible, I dunno it's a bit like if you were making sweets why should, you're not really gonna decide that you're gonna add less sugar or something like that I don't know I just, it doesn't seem necessarily like it's the right people to be making a decision on it when they're gonna be biased about it anyway.'*—Charlotte (PGSI score = 2)

Charlotte expresses concern over gambling operators including safer gambling content within their marketing, instead stating a preference for information to come from a less biased source. This indicates a lack of trust in gambling operators to provide useful safer gambling advice, since they stand to financially benefit from riskier gambling behaviour. It also implies that the effectiveness of messages used is further limited by the very fact they are viewed as a biased source. Essentially, if they do not believe that operators want their customers to gamble in a safe and controlled manner, then they will not follow any safer gambling advice given within their marketing.

Another way that safer gambling messages lacked effectiveness was the common misconception that safer gambling is a reactionary measure aimed at helping those who are addicted, rather than a general principle for all gamblers to follow. For example, Connor argues safer gambling adverts are ineffective as they won't help those who are addicted.

*'I think that people with the problem are the people that are addicted so you've got to tackle, I feel like the solution has got to be a bit more of a harder one I mean like yes you can relate to those things and but like just because a pundit said you shouldn't make a, put a bet on because you've lost the last, you're on a losing streak or whatever, it doesn't really make it, especially that one where it's saying don't bet when you're drunk but it, when you're drunk you've got less control anyways so whether that's at the forefront of your mind I'm not really sure, er, I'm not really sure that it works at all to be honest.'*—Connor (PGSI score = 6)

In framing their response to the question of the advert's effectiveness in relation to how it may impact individuals experiencing gambling problems, Connor implies that the purpose of these adverts is to get disordered gamblers to identify and change their problematic behaviour. As such, safer gambling messages are seen as irrelevant for those who do not identify their behaviour as problematic as they do not feel the need to change their behaviour.

## Discussion

### Contribution to existing theory and literature

The current findings supported a recently published interview study which also investigated sports bettors' perceptions of gambling marketing within Great Britain [15], in that marketing offers were seen to reduce risk. However, whilst that study explained how gambling promotions decreased feelings of risk associated with a particular bet, the current study highlights how gamblers feel as though they can carefully take advantage of marketing offers to reduce the overall risk associated with their gambling behaviour. A deeper interpretation of the data

highlighted how this reduction of risk appears to alleviate internal conflicts around the frequency of gambling behaviour. Such an internal conflict can be seen as an example of cognitive dissonance [40], whereby the frequency of their gambling behaviour may differ from their perception of what a 'safe' frequency of gambling is. Therefore, engaging with marketing offers to reduce risk may allow gamblers to employ a form of internal self-justification, whereby attitudes are altered to make negative consequences seem more tolerable and reduce states of cognitive dissonance [41].

It was also highlighted that participants viewed certain types of marketing as being free of risk and therefore allowing greater freedom to choose bets with longer odds. This finding is supported by experimental research that found that participants chose significantly larger odds when a betting incentive was offered [13]. Interestingly, it was suggested that free bet offers can lead to continued choices of longer odds bets due to emotional states extending beyond the use of the inducement. This may be explained by research which has demonstrated a relationship between the value of expected winnings and subjective measures of excitement, as well as increased heart rates [42]. Therefore, if individuals experience the excitement of larger potential winnings, this may encourage them to choose such bets again in the future. This highlights how the intended use of marketing offers may not always align with the outcomes of interacting with such offers. Whilst gamblers may think of marketing as a tool by which they can reduce their risk, it may lead to riskier behaviour over a longer time period due to the increased volatility of bets with higher odds.

A further way in which gamblers reported taking advantage of marketing within the study was through making judgements as to when offers increased the value of bets. One participant discussed how some offers can boost the odds of bets to odds which give a lower implied probability than his perceived likelihood of the bet winning. Such a finding aligns with a recent systematic review which concluded that sports bettors attributed more importance to skill than luck in the outcomes of betting [43]. The current study builds upon these findings to highlight how such perceptions of gambling as a skilled activity can impact how gamblers interact with marketing offers. This is a concern as the same review found that sports bettors perform no better with their choices of bets than random selection [43]. Additionally, sports bettors have been found to overestimate the probability of more complex bets often included in marketing offers [8]. So, whilst bettors may think of marketing as something which can be exploited for financial gain, this may not always be possible due to an overestimation of their own skill. Additionally, sports betting adverts use a dual persuasion strategy, to enhance perceived control and reduce perceived risk [44]. The findings of the current study suggest that such persuasion strategies are successful as they are reflected within the way gamblers think about gambling marketing.

Participants within the study acknowledged the risk associated with gambling marketing. A number of participants discussed how marketing drew them back into gambling after a period of abstinence, suggesting marketing acted as challenge to maintaining a change in behaviour. For example, one participant described how marketing reminded him of the enjoyment he would get from gambling after choosing to not gamble for a while and another discussed trying to actively avoid marketing during periods they were uncomfortable with their own gambling behaviour. Previous qualitative research found similar findings in that gambling adverts acted as a reminder to gamble and initiated gambling sessions [7, 45], however this finding this was mainly amongst treatment-seeking disordered gamblers whereas nobody in the current study had a diagnosis of gambling disorder. Taken together, these findings highlight how marketing is perceived to prevent sustained behaviour change across the spectrum of gambling-related harm.

One potential theoretical explanation as to why marketing may prevent gamblers from maintaining behaviour change relates to the role of self-efficacy in leading models of behaviour change, such as the Transtheoretical Model [46] and the Theory of Planned Behaviour [47]. Both models argue that, for behaviour change to be successful, individuals must believe themselves capable of maintaining such a change. Within the Transtheoretical Model specifically, it is argued that behaviour change is often unsuccessful when feelings of temptation outweigh an individual's confidence in their ability to maintain a behaviour. Therefore, seeing gambling marketing may increase temptation to gamble above an individual's level of self-efficacy, prompting them to start gambling again. This is particularly relevant given that research has shown that advertisements include perceived control enhancing content [44]. Self-efficacy may therefore be an inducement to gamble when the efficacy is linked to a gambling outcome yet can act as a preventative factor when efficacy is linked to the ability to avoid temptation.

Alternatively, the Theory of Planned Behaviour suggests that self-efficacy is just one important element of engaging in a particular behaviour. Subjective norms, the extent to which an individual believes others approve of a behaviour, and an individual's own attitudes towards a behaviour are also important in producing behavioural intentions. Given that gambling marketing has been highlighted as a major factor in normalising gambling within society [5], and that gambling is often presented positively in marketing, this could act to lower the desire to maintain changes in behaviour by increasing positive attitudes towards gambling. In addition to this, gambling in response to marketing may be considered a form of reminder impulse purchasing [48, 49] and a recent meta-analysis highlighted a link between positive emotions and increased impulse purchasing [50]. So, marketing which aims to increase positive feelings towards gambling may not only lower desire to maintain changes in gambling behaviour but also increase the temptation to engage within impulse purchasing.

However, despite acknowledging the risks associated with gambling marketing, participants described marketing as only being a serious problem for those with a diagnosis of gambling disorder or with a specific type of personality. An explanation for such an argument is the third-person effect, which refers to an individual's belief that mass media messages have a larger impact on others than themselves [51]. However, given that participants were keen to stress the rationality of their behaviour in comparison to the perceived vulnerabilities of specific types of people, their explanation moves beyond the third-person effect. By distancing themselves from any harm arising from gambling marketing, participants can protect a positive identity and stress rationality and self-control as markers of positive identity in contrast to disordered gamblers. One possible explanation for this is that it is an example of a fundamental attribution error [52], whereby individuals over-emphasise personality characteristics over situational explanations when explaining the behaviour of others.

Alternatively, if this distinction between their behaviour and others' behaviour is more intentional, this could be seen as being rooted in Social Identity Theory [53]. This theory argues individuals attribute negative characteristics to an out-group, in this case 'vulnerable gamblers', in order to enhance their own self-image. Such categorisation of gamblers into two distinct subgroups of 'safe' and 'vulnerable' gamblers can be seen as a reflection of the narrative around individual control which is often supported by the gambling industry [54]. This is a concern since gambling does not only harm those with a diagnosis of gambling disorder [12]. Also, research has found that certain aspects of gambling marketing led to riskier behaviour regardless of disordered gambling category [13]. So, despite thinking of marketing as something which only has a negative impact on others who cannot control their gambling, this may not accurately reflect the impact marketing has on gamblers' behaviour.

The issue of identity is also relevant within the finding that safer gambling messages incorporated into gambling marketing are perceived to be largely ineffective. One reason for this

was the perception of safer gambling as a reactionary measure aimed at helping those who are disordered gamblers. If gamblers are motivated to maintain a positive identity of being a safe gambler, then safer gambling messages are not going to be perceived as being relevant due to targeting the wrong identity. Participants also expressed concerns over the sincerity of industry-led safer gambling messages due to the concept of safer gambling not aligning with their financial interests as a business. There were also concerns around their sincerity due to these messages usually being included as an afterthought within advertisements of gambling products and are their infrequency in comparison to gambling adverts. It is argued that source credibility, which incorporates both trustworthiness and expertise, is a key factor in whether health messages are accepted by their target population [55]. Thus, if gamblers perceive that the gambling industry is not a trustworthy source to deliver safer gambling messages then they will not accept or process the content of the messages.

## Methodological considerations and evaluation

One key methodological consideration of using IPA is to purposively recruit a homogenous sample [19]. The current study achieved this by recruiting in the age range that most frequently report gambling in response to marketing [31] and who gambled frequently but did not have a diagnosis of gambling disorder. Such levels of homogeneity within the sample is a strength of the study and help addressed previously discussed limitations in the research area. This is important given the large number of individuals who meet the criteria for being at-risk gamblers within Great Britain [56]. However, an inherent limitation of such a sampling approach means you may not pick up on important experiences of the phenomena in other relevant demographics. For example, the sample recruited was largely white-British, male, undergraduate students and between the ages of 18 and 23. In particular, those aged 24 to 34 are also included in the age category of those who gamble most in response to gambling marketing and are likely to be more experienced gamblers than their younger counterparts. Therefore, the current study may have missed some valuable experiences of how these bettors think about gambling marketing. Further qualitative research should explore how other demographic groups think about gambling marketing.

A further criticism often made of IPA as a methodology is that is overly subjective and therefore unscientific [57]. However, such levels of interpretation are actively encouraged within IPA as a means of exploring feelings, emotions, and meanings [25]. Additionally, care was taken to ensure transparency around the analysis process through documenting the different stages of the analysis (https://osf.io/utzds/). Whilst the lack of member checking may be considered a limitation of the current study, credibility of the analysis was instead strengthened through a second researcher (GP), who was unfamiliar with gambling research, reviewing the analysis to ensure that findings were grounded in the experiences of participants.

## Future research suggestions

Given the perceived lack of effectiveness of current safer gambling strategies identified here, future research should explore the impact of new safer gambling strategies upon gambling behaviour. For example, concerns were highlighted about both the sincerity and placement of safer gambling messages within marketing. Therefore, future research should assess the impact of safer gambling messages from impartial sources and messages presented separately from gambling marketing. There has also been recent research showing how pictograms increased the perceived risk associated with the products and improved discrimination of risk between products [58]. Future studies should therefore test the impact of including these types of

images within gambling marketing to improve risk perception, however there should also be an assessment on whether this directly impacts upon gambling behaviour.

## Conclusion

The current study aimed to explore how frequent gamblers think about gambling marketing and the impact it has upon behaviour. Participants disclosed that marketing was something that they felt they could exploit for their own personal gain, either through increasing the value of bets or by reducing the risk associated with betting. However, they also acknowledged that marketing acted as a test of their own self-control by tempting them to bet in situations where they had not planned on doing so. Despite this, participants were keen to stress that marketing had little serious impact upon their behaviour and that marketing was only a risk factor for other people who were seen to be more vulnerable to developing disordered gambling. Finally, participants saw safer gambling messages within marketing as being ineffective due to the perceived insincerity of incorporating safer gambling content withing gambling advertisements. These findings highlight the need for improved safer gambling strategies that move beyond individual responsibility. A public health approach to reducing gambling-harm should focus on eliminating control-enhancing aspects of gambling marketing and products, such as free bets and money-back offers. There should also be a rethink on the use of safer gambling messaging within marketing as it requires an evaluation of perceived risk amongst gamblers. Therefore, such messages appear to miss their intended audience if perceived risk does not align with objective levels of risk.

## Acknowledgments

Firstly, we would like the thank the participants in this study for so kindly sharing their personal experiences during the interview process. We'd also like to extend thanks to our reviewers for their informative feedback throughout the peer-review process.

## Author Contributions

**Conceptualization:** Scott Houghton, Emma Casey, Andrew McNeill, Mark Moss.

**Formal analysis:** Scott Houghton, Georgia Punton.

**Funding acquisition:** Scott Houghton.

**Investigation:** Scott Houghton.

**Methodology:** Scott Houghton, Andrew McNeill, Mark Moss.

**Project administration:** Scott Houghton, Georgia Punton.

**Supervision:** Emma Casey, Andrew McNeill, Mark Moss.

**Visualization:** Scott Houghton, Georgia Punton.

**Writing – original draft:** Scott Houghton.

**Writing – review & editing:** Scott Houghton, Georgia Punton, Emma Casey, Andrew McNeill, Mark Moss.

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
