## [Decision Letter · Decision Letter 0]

3 Mar 2023

PONE-D-22-31833Frequent gamblers’ perceptions of the role of gambling marketing in their behaviour: an Interpretative Phenomenological AnalysisPLOS ONE

Dear Dr. Houghton,

Thank you for submitting your manuscript to PLOS ONE. After careful consideration, we feel that it has merit but does not fully meet PLOS ONE’s publication criteria as it currently stands. Therefore, we invite you to submit a revised version of the manuscript that addresses the points raised during the review process. The reviewers (especially 2 and 3) have done a very careful job, so I will not reiterate all their points. Still, please pay special attention to the methodological concerns expressed by reviewer 2. Please submit your revised manuscript by Apr 17 2023 11:59PM. If you will need more time than this to complete your revisions, please reply to this message or contact the journal office at plosone@plos.org. Please include the following items when submitting your revised manuscript:A rebuttal letter that responds to each point raised by the academic editor and reviewer(s). You should upload this letter as a separate file labeled 'Response to Reviewers'.A marked-up copy of your manuscript that highlights changes made to the original version. You should upload this as a separate file labeled 'Revised Manuscript with Track Changes'.An unmarked version of your revised paper without tracked changes. You should upload this as a separate file labeled 'Manuscript'.If applicable, we recommend that you deposit your laboratory protocols in protocols.io to enhance the reproducibility of your results. Protocols.io assigns your protocol its own identifier (DOI) so that it can be cited independently in the future. For instructions see: https://journals.plos.org/plosone/s/submission-guidelines#loc-laboratory-protocols. Additionally, PLOS ONE offers an option for publishing peer-reviewed Lab Protocol articles, which describe protocols hosted on protocols.io. Read more information on sharing protocols at https://plos.org/protocols?utm_medium=editorial-email&utm_source=authorletters&utm_campaign=protocols.

We look forward to receiving your revised manuscript.

Kind regards,

José C. Perales

Academic Editor

PLOS ONE

Journal Requirements:

Reviewers' comments:

Reviewer's Responses to Questions

**Comments to the Author**

1. Is the manuscript technically sound, and do the data support the conclusions?

Reviewer #1: Yes

Reviewer #2: Yes

Reviewer #3: Yes

2. Has the statistical analysis been performed appropriately and rigorously? 

Reviewer #1: N/A

Reviewer #2: N/A

Reviewer #3: I Don't Know

3. Have the authors made all data underlying the findings in their manuscript fully available?

Reviewer #1: Yes

Reviewer #2: Yes

Reviewer #3: Yes

4. Is the manuscript presented in an intelligible fashion and written in standard English?

Reviewer #1: Yes

Reviewer #2: Yes

Reviewer #3: Yes

5. Review Comments to the Author

Reviewer #1: Overall

This is an interesting and thorough manuscript on a topical issue. It’s generally well written (although overly wordy – see below), and is certainly well considered. It’s clear that the authors have invested considerable effort into producing an intelligent and thoughtful analysis and interpretation. It was a pleasure to read a qualitative analysis that provides such depth and insights. I have some suggestions for improvement – mostly minor. I do think the paper is too long, so have suggested some ways to trim it.

Introduction

The introduction was well argued. I understand that cross-sectional quantitative studies cannot determine the effects of gambling marketing, but they nevertheless have provided useful findings on its perceived effects, and associations with gambling and harmful gambling. So, I think that a short summary of key findings would be valuable, rather than dismissing them entirely. There are several reviews of research into gambling marketing that the authors could draw on to write a few sentences on this to better set the scene for the value of their IPA research. Also, longitudinal studies can help to untangle causal relationships between exposure to gambling marketing and gambling behaviour. One that provides relevant insights is:

Browne, M. et al. (2019). The impact of exposure to wagering advertisements and inducements on intended and actual betting expenditure: An ecological momentary assessment study. Journal of Behavioral Addictions, 8(1), 146-156.

The range of gambling research literature that the manuscript draws upon is fairly lean. Additional qualitative studies that might further inform the intro and/or discussion are:

McGee, D. (2020). On the normalisation of online sports gambling among young adult men in the UK: A public health perspective. Public Health, 184, 89-94.

Parke, A., & Parke, J. (2019). Transformation of sports betting into a rapid and continuous gambling activity: A grounded theoretical investigation of problem sports betting in online settings. International Journal of Mental Health and Addiction, 17(6), 1340-1359.

The interview chapter in this peer-reviewed report, which used a grounded theory methodology and highlights some similar themes to your analysis: Hing, N. et al. (2018). Effects of wagering marketing on vulnerable adults. Melbourne: Victorian Responsible Gambling Foundation. https://responsiblegambling.vic.gov.au/resources/publications/effects-of-wagering-marketing-on-vulnerable-adults-408/

I think Table 12 should be Table 1?

Methods

The authors provide an excellent justification for their use of IPA, and a very thorough explanation of the methods used. I appreciated having access to the interview guide and analytical processes. However, I’d encourage the authors to reduce the length of this section. This could be achieved by focusing on more concise phrasing where possible, eliminating any repetition, and removing less important information. For example, the info in the section on the Role of the researcher could be stated in a couple of sentences.

Results

The analysis provided a very thoughtful and impressive interpretation. I have no suggestions for any substantive changes.

Again however, some expression could be more concise (i.e., edit with the aim of reducing the length of each sentence and removing unnecessary words). Some quotes could also be trimmed by replacing words that don’t really add anything with “…”. Some quotes would benefit from more punctuation to improve their readability.

Discussion

I don’t think the Summary of results is needed since the intro to the Results section already provides this. Suggest removing.

The Discussion was very well considered and I appreciated the discussion of theoretical explanations of the findings. Again, I don’t think it requires any substantive changes.

As a point of accuracy, Hing et al. (2014) also recruited 50 online gamblers from the general population, and at varying levels of gambling severity, so it isn’t accurate to say they only recruited disordered gamblers.

The section on methodological considerations and evaluation could be much shorter, e.g., the benefits of IPA do not need repeating here. And much of what’s in the last paragraph of this section repeats what has already been explained.

The authors should acknowledge the limitations of the participants all being university students, all but one being White-British, and all being male except one.

Future research strategies could be stated much more concisely, in a sentence or two.

Conclusion

I’d like to see a broader consideration of implications beyond just better safer gambling messages/strategies – or at least more clarification of the potential scope of these strategies. The authors acknowledge that gambling harm is not restricted to those with a GD diagnosis, in recognition that gambling is a public health issue and not just an issue of individual pathology. So, I was surprised that the implications did not extend into areas beyond self-regulatory strategies for individual gamblers such as messages, to also include the need for safer gambling products, environments, regulation and marketing – in line with a public health approach. I’m not suggesting a detailed discussion is needed – but at least an acknowledgement that safer gambling strategies extend beyond messages (which in any case are a very weak measure), to also include changes to gambling products, environments, advertising, etc.

At minimum, it would be appropriate to comment on how gambling advertisements, inducements and other marketing strategies could be curtailed or otherwise regulated to pose a lesser risk to consumers.

Manuscript length

The manuscript is currently very long (around 9,000 words) and this unfortunately detracts from its overall impact. While PONE does not have a strict word length, they do “encourage you to present and discuss your findings concisely”.

While I understand that qualitative research seeks to provide in-depth insights, the expression itself in the manuscript tends to be overly wordy and long-winded. The authors could trim a lot of the length by tightening up expression to remove unnecessary words and lengthy phrasing. It’s surprising how much word length can be trimmed without losing valuable content, by an edit focused solely on achieving this.

Reviewer #2: In my opinion, the article under review is an appreciable approach to an area of gambling not yet studied in too much depth. That is, the narratives and experiences of active gamblers regarding the impact on their own behaviour of the advertising tactics employed by the gambling industry. Furthermore, the methodology employed is intended to complement experimental research and to guide policy decisions on regulating gambling marketing and 'responsible gambling' messages. However, certain weaknesses are found, mainly in the selection of the sample and the justification of the methodology used; which, in my opinion, would require corrections.

Therefore, my recommendation is that the weaknesses observed should be corrected or reformulated in order to face a subsequent evaluation that will decide on the eventual acceptance of the publication of this article in Plos One with greater robustness.

In terms of the most important issues, I specify below what should be corrected or improved in my view:

1. Although a laudable effort is made to credit the goodness and usefulness of Interpretative Phenomenological Analysis (IPA) in this study, I think there is one aspect that is not sufficiently justified. In the introduction, lines 100-101, there is talk of subtle marketing strategies: how can IPA bring out mechanisms during an interview that, by their nature, are beyond conscious control?

2. I miss a consistent justification of the sample size. At this point, some explanation of data reaching saturation would have been necessary (in fact, it is requested in the journal's guidelines for qualitative studies). In this sense, the following reference could be useful to the authors in addressing the problem: Boddy, C. R. (2016). Sample size for qualitative research. Qualitative Market Research: An International Journal, 19(4), 426-432.

3. Continuing with the choice of the sample, I infer that the resulting homogeneity in age and gender of the sample is due to the interviewer sharing these characteristics with the participants and 'this allowed a rapport to be developed that encouraged detailed experiential responses from participants' (lines 213-214). I believe that this claim would need to be supported by the available evidence. On the other hand, I note that, although one inclusion criterion is being in the 18-34 age range because 'as individuals from this group report gambling mostly in response to marketing' (lines 134-135), most of the group are 21-22 years old. Is this not missing valuable information from people who are slightly older and probably more experienced (23-34 years old)?

4. I would appreciate more details on informed consent: was there a possibility to leave the interview at any time without any penalty, were there people who did not sign the consent form and therefore did not participate in the study?

5. The journal's publication guidelines recommend following the checklist 'Consolidated criteria for reporting qualitative studies (COREQ)'. In the revised article, some points are missing, for example, the experience and training of the interviewer, in particular with the methodology used; the already mentioned data saturation; or whether the transcripts were reviewed by the participants to allow them for further comments or corrections.

6. I feel that there is a missing reference to support this statement: ‘However, assessing the impact of marketing upon gambling behaviour proves to be methodologically challenging due to the wide-ranging marketing strategies employed’ (lines 50-52).

7. I find that in the section 'Methodological considerations and evaluation' little space is given to the limitations of the methodology and the study. I think that these limitations could be explored a little more in depth. On the other hand, mention is made of a subject whose information seems to be of poor quality. Has the possibility of eliminating him/her from the sample been considered? And, if kept, with what justification?

Minor issues would be the following:

8. I find that several doi links are broken, check.

9. Check the in-text citations and/or references of Lopez-Gonzalez, Gambling Comission, Ferris & Wynne, Pietkiewicz et al., Killick, Holland and Hocevar.

10. On line 276, James does not have a severity score of 1.

11. According to the publication guidelines, the article should contain an acknowledgements section.

12. Also according to the publication guide: ‘PLOS uses the reference style outlined by the International Committee of Medical Journal Editors (ICMJE), also referred to as the “Vancouver” style’.

Finally, I would like to congratulate the authors for their efforts and especially note their commitment to transparency in science by making such a wealth of valuable information available to other researchers.

I hope that my comments make sense and serve to improve the text. And I confirm my availability to re-examine the revised version.

Reviewer #3: I consider this qualitative study to be a relevant contribution to our understanding of the subjective impact of gambling marketing on consumers. I would like to highlight the transparency of the method and the public accessibility of the results. I also think that this study has several strengths, such as the way it is written and the comprehensive theoretical background. However, and although it is not a serious problem for my general positive opinion of the work, I wonder why the authors did not use some psychometric instruments as a way to support qualitative analyses. I also think that the generalizability of the data could be problematic since the study is carried out with university students. I miss a greater emphasis on this limitation. Finally, I wonder if it is possible to better relate these findings to the gambling harm prevention literature focused on gambling products, along the lines, for example, of the recent study by Luquiens et al, 2022. In my opinion, this study also supports the idea of protecting vulnerable users from the pernicious effect of some advertising strategies.

6. PLOS authors have the option to publish the peer review history of their article (what does this mean?). If published, this will include your full peer review and any attached files.

Reviewer #1: **Yes: **Nerilee Hing

Reviewer #2: **Yes: **Jose López-Guerrero

Reviewer #3: No

---

## [Author Response · Author response to Decision Letter 0]

22 Mar 2023

***These responses are also included in an attached response letter***

Journal Requirements:

Response: We have edited the manuscript formatting to align with the requirements stated in both of the links provided. We have also conducted a full review of our reference list and fixed some issues with DOI links that were not working, as well as switching to the required Vancouver referencing style. There were no retracted articles within the reference list.

Reviewers' comments:

Reviewer #1: Overall

This is an interesting and thorough manuscript on a topical issue. It’s generally well written (although overly wordy – see below), and is certainly well considered. It’s clear that the authors have invested considerable effort into producing an intelligent and thoughtful analysis and interpretation. It was a pleasure to read a qualitative analysis that provides such depth and insights. I have some suggestions for improvement – mostly minor. I do think the paper is too long, so have suggested some ways to trim it.

Introduction

The introduction was well argued. I understand that cross-sectional quantitative studies cannot determine the effects of gambling marketing, but they nevertheless have provided useful findings on its perceived effects, and associations with gambling and harmful gambling. So, I think that a short summary of key findings would be valuable, rather than dismissing them entirely. There are several reviews of research into gambling marketing that the authors could draw on to write a few sentences on this to better set the scene for the value of their IPA research. Also, longitudinal studies can help to untangle causal relationships between exposure to gambling marketing and gambling behaviour. One that provides relevant insights is:

Browne, M. et al. (2019). The impact of exposure to wagering advertisements and inducements on intended and actual betting expenditure: An ecological momentary assessment study. Journal of Behavioral Addictions, 8(1), 146-156.

The range of gambling research literature that the manuscript draws upon is fairly lean. Additional qualitative studies that might further inform the intro and/or discussion are:

McGee, D. (2020). On the normalisation of online sports gambling among young adult men in the UK: A public health perspective. Public Health, 184, 89-94.

Parke, A., & Parke, J. (2019). Transformation of sports betting into a rapid and continuous gambling activity: A grounded theoretical investigation of problem sports betting in online settings. International Journal of Mental Health and Addiction, 17(6), 1340-1359.

The interview chapter in this peer-reviewed report, which used a grounded theory methodology and highlights some similar themes to your analysis: Hing, N. et al. (2018). Effects of wagering marketing on vulnerable adults. Melbourne: Victorian Responsible Gambling Foundation. https://responsiblegambling.vic.gov.au/resources/publications/effects-of-wagering-marketing-on-vulnerable-adults-408/

Response: Thank you for the feedback provided here. I’ve added some information on other key findings of existing quantitative research at the start of the 2nd paragraph. I’ve also discussed the promising nature of the findings of the linked longitudinal study. I’ve also discussed the findings of the grounded theory in both the introduction and discussion – this was very useful as I have not seen this before and the idea of minimising losses being a core component of response to marketing aligns very well with the findings of this study.

I think Table 12 should be Table 1?

Response: Thank you for highlighting this, it has now been changed.

Methods

The authors provide an excellent justification for their use of IPA, and a very thorough explanation of the methods used. I appreciated having access to the interview guide and analytical processes. However, I’d encourage the authors to reduce the length of this section. This could be achieved by focusing on more concise phrasing where possible, eliminating any repetition, and removing less important information. For example, the info in the section on the Role of the researcher could be stated in a couple of sentences.

Response: Thank you for your kind comments. I have attempted to make my wording more concise and I have greatly reduced the length of the ‘role of the researcher’ section.

Results

The analysis provided a very thoughtful and impressive interpretation. I have no suggestions for any substantive changes.

Again however, some expression could be more concise (i.e., edit with the aim of reducing the length of each sentence and removing unnecessary words). Some quotes could also be trimmed by replacing words that don’t really add anything with “…”. Some quotes would benefit from more punctuation to improve their readability.

Response: Thank you for the kind words. We have again tried to make the writing style more concise and shortened some of the quotes where possible.

Discussion

I don’t think the Summary of results is needed since the intro to the Results section already provides this. Suggest removing.

The Discussion was very well considered and I appreciated the discussion of theoretical explanations of the findings. Again, I don’t think it requires any substantive changes.

As a point of accuracy, Hing et al. (2014) also recruited 50 online gamblers from the general population, and at varying levels of gambling severity, so it isn’t accurate to say they only recruited disordered gamblers.

The section on methodological considerations and evaluation could be much shorter, e.g., the benefits of IPA do not need repeating here. And much of what’s in the last paragraph of this section repeats what has already been explained.

The authors should acknowledge the limitations of the participants all being university students, all but one being White-British, and all being male except one.

Future research strategies could be stated much more concisely, in a sentence or two.

Response: Thank you for all of these suggestions. We have removed the summary of results as suggested. I have also edited the sentence relating to the Hing et al (2014) article to make it more clear that I meant the finding of marketing acting as temptation for gambling in those who wanted to reduce their behaviour was mostly in participants who were treatment-seeking gamblers (rather than implying that the studies only recruited treatment-seeking gamblers). I’ve rewritten the methodological considerations section – it is now a lot more concise and have noted the identified issues with the sample. I’ve also made the future research strategies more concise.

Conclusion

I’d like to see a broader consideration of implications beyond just better safer gambling messages/strategies – or at least more clarification of the potential scope of these strategies. The authors acknowledge that gambling harm is not restricted to those with a GD diagnosis, in recognition that gambling is a public health issue and not just an issue of individual pathology. So, I was surprised that the implications did not extend into areas beyond self-regulatory strategies for individual gamblers such as messages, to also include the need for safer gambling products, environments, regulation and marketing – in line with a public health approach. I’m not suggesting a detailed discussion is needed – but at least an acknowledgement that safer gambling strategies extend beyond messages (which in any case are a very weak measure), to also include changes to gambling products, environments, advertising, etc.

At minimum, it would be appropriate to comment on how gambling advertisements, inducements and other marketing strategies could be curtailed or otherwise regulated to pose a lesser risk to consumers.

Response: Thank you for suggesting this, I have now edited the conclusion to address the wider-reaching implications of the findings, such as the need for more of a public-health approach to reducing harm and to rethink safer gambling messaging. 

Manuscript length

The manuscript is currently very long (around 9,000 words) and this unfortunately detracts from its overall impact. While PONE does not have a strict word length, they do “encourage you to present and discuss your findings concisely”.

While I understand that qualitative research seeks to provide in-depth insights, the expression itself in the manuscript tends to be overly wordy and long-winded. The authors could trim a lot of the length by tightening up expression to remove unnecessary words and lengthy phrasing. It’s surprising how much word length can be trimmed without losing valuable content, by an edit focused solely on achieving this.

Response: Thank you for highlighting this. I have managed to reduce the word count to 7,870 words. This was achieved through removing some sections as suggested within your review, cutting down on some of the quotes in the results section and trying to remove any unnecessary wording or phrasing. This was offset a little bit by making additions suggested by yourself and reviewer 2, however I think the changes overall should have made for a stronger and more concise manuscript.

Reviewer #2:

In my opinion, the article under review is an appreciable approach to an area of gambling not yet studied in too much depth. That is, the narratives and experiences of active gamblers regarding the impact on their own behaviour of the advertising tactics employed by the gambling industry. Furthermore, the methodology employed is intended to complement experimental research and to guide policy decisions on regulating gambling marketing and 'responsible gambling' messages. However, certain weaknesses are found, mainly in the selection of the sample and the justification of the methodology used; which, in my opinion, would require corrections.

Therefore, my recommendation is that the weaknesses observed should be corrected or reformulated in order to face a subsequent evaluation that will decide on the eventual acceptance of the publication of this article in Plos One with greater robustness.

In terms of the most important issues, I specify below what should be corrected or improved in my view:

1.Although a laudable effort is made to credit the goodness and usefulness of Interpretative Phenomenological Analysis (IPA) in this study, I think there is one aspect that is not sufficiently justified. In the introduction, lines 100-101, there is talk of subtle marketing strategies: how can IPA bring out mechanisms during an interview that, by their nature, are beyond conscious control?

Response: Thank you for highlighting the lack of clarity here. Instead of attempting to understand the objective “impact” of these more subtle marketing strategies, the use of IPA methodology can encourage reflection upon these strategies to understand how gamblers perceive and engage with such strategies. Some marketing strategies involve the exploitation of cognitive biases, but even though the bias itself is beyond the consciousness of the gambler, the actual perception that one set of odds looks more favourable than another set of odds (because of how the odds are presented) is subject of conscious awareness. Thus, the bias itself is subliminal, but the effect of the bias produces a reflection (about the weight of the odds) that is phenomenological. This has now been made more clear within the manuscript.

2. I miss a consistent justification of the sample size. At this point, some explanation of data reaching saturation would have been necessary (in fact, it is requested in the journal's guidelines for qualitative studies). In this sense, the following reference could be useful to the authors in addressing the problem: Boddy, C. R. (2016). Sample size for qualitative research. Qualitative Market Research: An International Journal, 19(4), 426-432.

Response: Thank you for highlighting the lack of justification present in the manuscript. The idea of data saturation is not a goal of IPA research – as the focus is instead placed on obtain rich and personal accounts for those that are recruited. It can also be argued that data saturation can never truly be reached when taking such an idiographic approach, as everyone’s experiences are so unique that there’s always the potential for new information to be brought up. Instead, the sample size was guided by recommendations from leading researchers in the area of between 4-10 participants for doctoral research (with more of a focus on recruiting people who can offer detailed insight into the phenomena studied).

3. Continuing with the choice of the sample, I infer that the resulting homogeneity in age and gender of the sample is due to the interviewer sharing these characteristics with the participants and 'this allowed a rapport to be developed that encouraged detailed experiential responses from participants' (lines 213-214). I believe that this claim would need to be supported by the available evidence. On the other hand, I note that, although one inclusion criterion is being in the 18-34 age range because 'as individuals from this group report gambling mostly in response to marketing' (lines 134-135), most of the group are 21-22 years old. Is this not missing valuable information from people who are slightly older and probably more experienced (23-34 years old)?

Response: Thank you for highlighting this. I have added a reference that discusses the benefits of such peer-interviewing within qualitative research. I think your argument on the age of the sample is a fair one that was not given appropriate coverage within the first draft of the manuscript. We have now acknowledged this limitation in the revised “methodological considerations and evaluations” sub-section.

4. I would appreciate more details on informed consent: was there a possibility to leave the interview at any time without any penalty, were there people who did not sign the consent form and therefore did not participate in the study?

Response: I have added this information into the data collection section.

5. The journal's publication guidelines recommend following the checklist 'Consolidated criteria for reporting qualitative studies (COREQ)'. In the revised article, some points are missing, for example, the experience and training of the interviewer, in particular with the methodology used; the already mentioned data saturation; or whether the transcripts were reviewed by the participants to allow them for further comments or corrections.

Response: Thank you for highlight this. The COREQ has now been updated in relation to the points mentioned here. I have explained that I received previous training in conducting IPA research during my postgraduate degree, have explained my stance on data saturation/sample size that I covered in response to point 2 and have explained that whilst member checking was not conducted, a 2nd researcher reviewed the analysis to ensure that the findings were grounded in participants experiences (thus improving the credibility of the analysis). The lack of member checking has also been acknowledged in the manuscript as a limitation in the discussion section.

6. I feel that there is a missing reference to support this statement: ‘However, assessing the impact of marketing upon gambling behaviour proves to be methodologically challenging due to the wide-ranging marketing strategies employed’ (lines 50-52).

Response: Thank you for highlighting this. I have now included a reference to the Newall (2019) review of gambling marketing research which covers numerous methodological challenges of assessing the impact of gambling marketing.

7. I find that in the section 'Methodological considerations and evaluation' little space is given to the limitations of the methodology and the study. I think that these limitations could be explored a little more in depth. On the other hand, mention is made of a subject whose information seems to be of poor quality. Has the possibility of eliminating him/her from the sample been considered? And, if kept, with what justification?

Response: This section has now been rewritten and we have attempted to address the balance between considering the strengths and limitations of the methodology/approach. We understand why making the argument about the general poor quality of data from 1 participant was confusing and we have removed it from the manuscript. For clarification purposes, the participant (Charlotte) did still offer some interesting perspectives (e.g. contributions to the results section on safer gambling) and so that is why they were retained, however the general quality and depth of her data was not at the same level as other participants.

Minor issues would be the following:

8. I find that several doi links are broken, check.

9. Check the in-text citations and/or references of Lopez-Gonzalez, Gambling Comission, Ferris & Wynne, Pietkiewicz et al., Killick, Holland and Hocevar.

10. On line 276, James does not have a severity score of 1.

11. According to the publication guidelines, the article should contain an acknowledgements section.

12. Also according to the publication guide: ‘PLOS uses the reference style outlined by the International Committee of Medical Journal Editors (ICMJE), also referred to as the “Vancouver” style’.

Finally, I would like to congratulate the authors for their efforts and especially note their commitment to transparency in science by making such a wealth of valuable information available to other researchers.

I hope that my comments make sense and serve to improve the text. And I confirm my availability to re-examine the revised version.

Response: Thank you for insightful feedback and highlighting these issues [8-12]. The correct referencing style has now been used, all references have been checked and all DOIs are now working as of the time of resubmitting the article. We have included an acknowledgements section and edited the notified error on line 278 [point 10].

Reviewer #3:

 I consider this qualitative study to be a relevant contribution to our understanding of the subjective impact of gambling marketing on consumers. I would like to highlight the transparency of the method and the public accessibility of the results. I also think that this study has several strengths, such as the way it is written and the comprehensive theoretical background. However, and although it is not a serious problem for my general positive opinion of the work, I wonder why the authors did not use some psychometric instruments as a way to support qualitative analyses. I also think that the generalizability of the data could be problematic since the study is carried out with university students. I miss a greater emphasis on this limitation. Finally, I wonder if it is possible to better relate these findings to the gambling harm prevention literature focused on gambling products, along the lines, for example, of the recent study by Luquiens et al, 2022. In my opinion, this study also supports the idea of protecting vulnerable users from the pernicious effect of some advertising strategies.

Response: Thank you for your kind comments on the manuscript. We did use the Problem Gambling Severity Index to assess the risk level of each participants in the study – however this was really just to contextualise the extent of their gambling behaviour. We have reworked the methodological limitations section to now include a larger focus on the sample recruited, where we discuss the benefits and limitations of recruiting such a homogeneous sample. Thank you for highlighting the Luquins paper, I think their findings accompany the ‘perceived risk’ finding of the current study and we have adjusted the future research suggestions to assess how including such pictograms within marketing may impact upon risk perception and gambling behaviour.

---

## [Decision Letter · Decision Letter 1]

5 Jun 2023

Frequent gamblers’ perceptions of the role of gambling marketing in their behaviour: an Interpretative Phenomenological Analysis

PONE-D-22-31833R1

Dear Dr. Houghton,

We’re pleased to inform you that your manuscript has been judged scientifically suitable for publication and will be formally accepted for publication once it meets all outstanding technical requirements.

Kind regards,

José C. Perales

Academic Editor

PLOS ONE

Additional Editor Comments (optional):

Reviewers' comments:

Reviewer's Responses to Questions

**Comments to the Author**

1. If the authors have adequately addressed your comments raised in a previous round of review and you feel that this manuscript is now acceptable for publication, you may indicate that here to bypass the “Comments to the Author” section, enter your conflict of interest statement in the “Confidential to Editor” section, and submit your "Accept" recommendation.

Reviewer #1: All comments have been addressed

Reviewer #2: All comments have been addressed

2. Is the manuscript technically sound, and do the data support the conclusions?

Reviewer #1: Yes

Reviewer #2: Yes

3. Has the statistical analysis been performed appropriately and rigorously? 

Reviewer #1: N/A

Reviewer #2: N/A

4. Have the authors made all data underlying the findings in their manuscript fully available?

Reviewer #1: Yes

Reviewer #2: Yes

5. Is the manuscript presented in an intelligible fashion and written in standard English?

Reviewer #1: Yes

Reviewer #2: Yes

6. Review Comments to the Author

Reviewer #1: I would like to thank the authors for thoroughly addressing my suggested changes. The manuscript makes a valuable contribution to the literature and I recommend its publication. Well done to the authors on a rigorous and insightful study.

Reviewer #2: In my opinion, the authors have satisfactorily addressed the requirements raised. My main concern was in relation to the justification of the sample size and whether data saturation had been reached. I believe that the authors have convincingly argued their rationale and this has been reflected in the manuscript. I would just like to add that the use of data saturation in phenomenological studies is an open debate: https://link.springer.com/article/10.1007/s11135-017-0574-8. Personally, it has helped me to learn about conflicting or complementary positions in the qualitative methodology area.

It only remains for me to congratulate the authors for this work, their commitment to the open science paradigm, and to wish them a fruitful future in the field of problem gambling.

7. PLOS authors have the option to publish the peer review history of their article (what does this mean?). If published, this will include your full peer review and any attached files.

Reviewer #1: **Yes: **Nerilee Hing

Reviewer #2: **Yes: **Jose López-Guerrero

---

## [Editor Report · Acceptance letter]

8 Jun 2023

PONE-D-22-31833R1 

Frequent gamblers’ perceptions of the role of gambling marketing in their behaviour: an interpretative phenomenological analysis 

Dear Dr. Houghton:

I'm pleased to inform you that your manuscript has been deemed suitable for publication in PLOS ONE. Congratulations! Your manuscript is now with our production department. 

Kind regards, 

on behalf of

Dr. José C. Perales 

Academic Editor

PLOS ONE